# Thresholds for clinical practice that directly link handgrip strength to remaining years of life: estimates based on longitudinal observational data

Sergei Scherbov [1], Sonja Spitzer [2], Nadia Steiber [3,4]

[1]International Institute for Applied Systems Analysis (IIASA), Wittgenstein Centre for Demography and Global Human Capital (IIASA, OeAW, University of Vienna), Laxenburg, Austria
[2]Department of Demography, University of Vienna, Wittgenstein Centre for Demography and Global Human Capital (IIASA, OeAW, University of Vienna), Vienna, Austria
[3]Department of Sociology, University of Vienna, Vienna, Austria
[4]Institute for Advanced Studies, Vienna, Austria

**Correspondence to**
Nadia Steiber;
nadia.steiber@univie.ac.at

## ABSTRACT

**Objective** Muscle strength is a powerful predictor of mortality that can quickly and inexpensively be assessed by measuring handgrip strength (HGS). What is missing for clinical practice, however, are empirically meaningful cut-off points that apply to the general population and that consider the correlation of HGS with gender and body height as well as the decline in HGS during processes of *normal* ageing. This study provides standardised thresholds that directly link HGS to remaining life expectancy (RLE), thus enabling practitioners to detect patients with an increased mortality risk early on.

**Design** Relying on representative observational data from the Health and Retirement Study, the HGS of survey participants was z-standardised by gender, age and body height. We defined six HGS groups based on cut-off points in SD; we use these as predictors in survival analyses with a 9-year follow-up and provide RLE by gender based on a Gompertz model for each HGS group.

**Participants** 8156 US American women and men aged 50–80 years.

**Main outcome measures** Z-standardised HGS and all-cause mortality.

**Results** Even slight negative deviations in HGS from the reference group with [0.0 SD, 0.5 SD) have substantial effects on survival. RLE among individuals aged 60 years with standardised HGS of [−0.5 SD, 0.0 SD) is 3.0/1.4 years lower for men/women than for the reference group, increasing to a difference of 4.1/2.6 years in the group with HGS of [−1.0 SD, −0.5 SD). By contrast, we find no benefit of strong HGS related to survival.

**Conclusions** HGS varies substantially with gender, age and body height. This confirms the importance of considering these heterogeneities when defining reference groups and risk thresholds. Moreover, survival appears to decrease at much higher levels of muscle strength than is assumed in previous literature, suggesting that medical practitioners should start to become concerned when HGS is slightly below that of the reference group.

## INTRODUCTION

Muscle strength, as frequently measured by handgrip strength (HGS), is a well-established indicator of functional status that has been widely used as a key component of frailty phenotypes[1][2] and in

## STRENGTHS AND LIMITATIONS OF THIS STUDY

⇒ We develop a clinical definition of low handgrip strength (HGS) related to increased mortality risk and provide cut-off points, along with straightforward visualisations, that can be used in medical practice.

⇒ In addition to HRs, we present, for the first time, estimates of remaining life expectancy related to the cut-off points, enabling a much more intuitive interpretation of mortality risk that can easily be communicated to patients.

⇒ Previous work suggests substantial differences in reference handgrip strength by race. This study provides cut-offs based on data from Caucasian individuals only, due to the small number of observations of HGS from other races.

diagnosing sarcopenia.[3][4] Beyond the use of HGS measurement in gerontological assessments, there is a large and growing body of research concerned with its prognostic value for future health conditions,[5] indicating that HGS is a powerful predictor of adverse health outcomes such as disability, cognitive decline and eventually mortality.[6–8]

Given its strong association with the functional status of older individuals and its high prognostic value for future morbidity and mortality, HGS has been promoted as a biomarker of healthy ageing. It has been shown to be a better predictor of survival and the maintenance of good health than chronological age[9] and can be used as a screening tool for the vulnerability of older individuals that is much simpler and more cost-effective than comprehensive geriatric assessments.[10][11]

While there is consensus that HGS could be used routinely to screen for people whose future health is at risk and who may require intervention, research on empirically relevant threshold values for such screenings is scarce.[12] Recent studies on the predictive

BMJ

value of HGS for mortality have used either a continuous measure of HGS in kilograms (kg), or broad and arbitrary classifications of HGS into quantiles.[6 12–15] Moreover, studies are often based on small, non-representative samples that are targeted at patients suffering from certain diseases only.[16 17] Some studies use mediating health outcomes to define thresholds. The Foundation of the National Institutes of Health,[18] for example, constructed cut-off points (in kg) based on the statistical associations between HGS and low gait speed, which were used as mortality predictors in a subsequent study.[19]

The most popular threshold for weak HGS—also frequently used today to predict survival—is not actually conceptualised with mortality in mind but, instead, for the diagnosis of sarcopenia. In 2010, the European Working Group on Sarcopenia in Older Persons (EWGSOP) recommended a cut-off at 2 SD below the mean value of HGS taken from a normative, healthy reference population to define muscle weakness.[3] In 2019, the group updated their thresholds, now suggesting a cut-off at 2.5 SD to define low HGS, translating to<27 kg for men and<16 kg for women.[4] This simple suggestion has become a common criterion, used worldwide, for the diagnosis of sarcopenia in geriatric assessment and gerontological research. Moreover, it is used in numerous studies as a threshold to predict mortality differentials.

However, when thresholds are defined as 2 or 2.5 SD below the mean value of a healthy reference population (ie, the t-scores), the decline in HGS during processes of *normal* ageing from age 40 years onwards[20]—which is associated with changing levels of physical activity, anabolic responsiveness to protein intake and hormonal status[21]—cannot be accounted for. This decline is not indicative of pathological ageing; in other words, it is not associated with mortality risks higher than those seen in other individuals of the same chronological age. Using a threshold based on t-scores would imply that the majority of older individuals are at a higher risk of dying and would simply reflect a lower life expectancy on the part of the old rather than provide an effective tool to detect the most vulnerable within each age group. Moreover, HGS increases substantially with body height,[20] which is an indicator neither of the absence of sarcopenia nor of higher survival. Adjusting HGS for body height thus has the potential to improve mortality risk assessments.[14] In a similar vein, women have a lower average HGS than men,[22] and HGS varies by race.[23]

In their most recent report, EWGSOP thus highlights the need for studies establishing gender-specific and region-specific threshold values to improve outcome prediction.[4] Going one step further and accounting for processes of *normal* ageing and the association of HGS with anthropometric traits, we propose standardised threshold values that consider the inherent, non-informative variations in muscle strength with gender, age and body height, that is, z-scores, using separate reference categories for each gender–age–height

group. The use of these threshold values enables practitioners to detect patients who deviate from the norm (ie, their reference group) and who would thus have a higher, or potentially lower, risk of death.

The aim of the present study is thus to define standardised threshold values for HGS that are associated with a substantially increasing risk of mortality, using z-scores. We provide estimates based on representative data for US Americans aged 50 years and above, irrespective of their health status, and directly link muscle strength to mortality, without any deviations to mediating diseases. In addition to defining standardised thresholds, we go beyond previous work by providing estimates of remaining life expectancy (RLE) associated with each of the cut-off points. In contrast to HRs, RLE provides information about actual life expectancy differences between HGS groups and can be communicated to patients more intuitively. Moreover, we produce clear illustrations of the relationship between tested HGS and the risk groups, something that can routinely be used in medical practice as a low-entry starting point for further geriatric assessments and health-enhancing patient interventions.

## METHODS
### Data, sample and study design
Our analysis is based on the Health and Retirement Study (HRS), a longitudinal panel survey providing representative data for the US American population aged 50 years and older.[24] In 2006, half the survey participants were randomly selected to perform an HGS test, with the other half completing the test in 2008. HRS can be linked to mortality register data from the National Death Index, and this allowed us to follow survey participants until December 2014. Participants can be linked to the National Death Index even if they leave the study early on, making attrition of minor concern for this analysis.

We restrict our sample to individuals aged 50–80 years, keeping all observations of HGS of at least 6 kg. To remove outliers in terms of anthropometric traits, we only keep women with a body height of 130–190 cm and men who are at least 150 cm in height. We focus on Caucasian individuals, as previous work shows a need for separate HGS thresholds by race,[23] and the number of observations for the other ethnic groups was too small for heterogeneity analyses. Finally, we drop cases with residuals larger than 2.5 SD from gender-specific linear regressions of age and body height on HGS (see next subsection for details). This leaves us with a sample of 8156 individuals to predict death with a follow-up period of up to 9 years, during which 978 participants died. More detailed sample characteristics are provided in table 1, and a chart visualising the sample construction can be found in online supplemental figure A1.

### Table 1  Participant characteristics stratified by gender

|  | Men | Women |
|---|---|---|
| Grip strength, kg, mean (SD) | 42.1 (8.4) | 25.5 (5.5) |
| Death during follow-up | 529 (14.8%) | 449 (9.8%) |
| Age, years, median (IQR) | 68.2 (61.8–73.3) | 67.7 (61.6–73.1) |
| Height, cm, median (IQR) | 174.6 (170.2–179.1) | 160.0 (156.2–164.5) |
| N | 3583 | 4573 |

IQR (interquartile range) 75%–25%.
SD, standard deviation.

## HGS measurement and z-standardisation by gender, age and body height

HGS was measured using a Smedley spring-type hand dynamometer. Two measurements were taken from each hand. Following published recommendations,[25] we used the maximum value achieved with either hand for our analyses.

As mentioned previously, there is a strong variation in HGS by gender, age and body height that is not directly linked to differences in mortality risks. We thus use a straightforward regression technique to adjust HGS values for these characteristics. First, we regress HGS on age and body height, using separate linear models for women (f) and men (m). Squared age and height terms were not significant. Regression analyses confirm the strong differences in HGS by gender, age and body height. More specifically, the expected HGS at baseline in 2006/2008 at ages 50–80 years can be expressed as:

$$HGS_m = 7.23 + 0.38 \times height(cm) - 0.46 \times age(yr) \ (adjusted \ R^2 : 0.28)$$

$$HGS_f = 6.00 + 0.25 \times height(cm) - 0.30 \times age(yr) \ (adjusted \ R^2 : 0.27)$$

Second, we calculate the standardised residuals from the previous equations, applying a z-standardisation with a mean of 0 and SD of 1 to obtain a measurement of HGS for each observation that is standardised for gender, age and body height, denoted in the following as $st\_HGS_i$. This allows for the construction of reference categories for each gender–age–height combination. Deviations in $st\_HGS_i$ from these reference categories enable mortality risk differentials to be detected.

In a third step, we define thresholds that allow for early detection of patients with an increased mortality risk. Attempts at defining optimal cut-off points using a log-rank or similar method produced non-robust results due to the monotonic increase in the mortality risk with $st\_HGS_i$. We thus construct six groups based on SD thresholds, as shown in table 2, rows 1 and 2. Given the high prevalence of chronic diseases in the study population, which is aged 50 years and above, we defined the 'healthy' reference group as those with $st\_HGS_i$ at or slightly above the mean [0.0 SD, 0.5 SD). The comparison groups comprise those with up to 0.5 SD below the mean (weak 1), those with less than a half to a full SD below the mean (weak 2), those with less than 1–2 SD below the mean (weak 3), those with less than 2 SD below the mean (weak 4) and those stronger than the reference group (strong). In the male sample, mean strength within the reference group was 43.9 kg, corresponding to about 3.9 kg more than the first weak group and about 17.1 kg more than the weakest group. In the female sample, mean strength within the reference group was 26.8 kg, corresponding to about 2.7 kg, more than the first weak group and about 12.0 kg more than the weakest group. Notably, the median age hardly varies between the HGS groups, given the standardisation.

### Table 2  HRs and sample characteristics by gender and st_HGS group

| Groups | Strong | Reference | Weak 1 | Weak 2 | Weak 3 | Weak 4 |
|---|---|---|---|---|---|---|
| Thresholds | [0.5 to 3.0) | [0.0 to 0.5) | [−0.5 to 0.0) | [−1.0 to −0.5) | [−2.0 to −1.0) | [−3.0 to −2.0) |
| **Men** | | | | | | |
| Median age | 68.2 | 67.9 | 68.8 | 68.5 | 67.6 | 66.2 |
| Mean grip in kg (SD) | 50.2 (5.8) | 43.9 (4.5) | 40.0 (4.6) | 36.5 (4.7) | 32.6 (4.7) | 26.8 (4.6) |
| HR (95% CI) | 0.93 (0.69 to 1.25) | 1.00 | 1.67 (1.23 to 2.26) | 2.02 (1.49 to 2.75) | 2.40 (1.77 to 3.26) | 2.34 (1.40 to 3.93) |
| N | 1142 (31.9%) | 688 (19.2%) | 647 (18.1%) | 522 (14.6%) | 481 (13.4%) | 103 (2.9%) |
| **Women** | | | | | | |
| Median age | 67.8 | 67.6 | 68.2 | 67.9 | 66.8 | 67.6 |
| Mean grip in kg (SD) | 30.8 (3.9) | 26.8 (2.9) | 24.1 (2.9) | 21.9 (2.9) | 19.1 (3.1) | 14.8 (2.9) |
| HR (95% CI) | 0.90 (0.66 to 1.22) | 1.00 | 1.32 (0.96 to 1.82) | 1.65 (1.20 to 2.28) | 1.85 (1.34 to 2.55) | 3.03 (0.83 to 5.04) |
| N | 1438 (31.4%) | 901 (19.7%) | 791 (17.3%) | 695 (15.2%) | 645 (14.1%) | 103 (2.3%) |

Extended tables with years of education as a control are provided in online supplemental table A1.
CI, confidence interval; HGS, handgrip strength; n, number of observations; SD, standard deviation.

## Statistical analysis

As a first step, we estimate the association between the six st_HGS groups (strong, reference, weak 1–4) and all-cause mortality, using separate Cox proportional hazard models for women and men. We control for age in the first model and add years of education for robustness analyses in a second model to account for the strong correlation of education with the speed of ageing.[26] As the estimated HRs are not informative in terms of differences in life expectancy, in a second step, we provide RLE estimates at age 60 and 70 years for both genders. More specifically, we approximate RLE by estimating segmented life expectancy from age 60 to 90 years and from age 70 to 90 years based on a Gompertz model that includes age as a covariate. Hence, strictly speaking, RLE gives the average number of years a person lives between the ages 60 and 90 years, as opposed to after 60 years. This approach is necessary because of the small number of people living beyond the age of 90 years in the cohorts observed and in the HRS sample—excluding them helps prevent biased estimates.

We applied bootstrapping, running 1000 simulations for each gender, and report the median RLE for each of the six st_HGS groups. Finally, we provide straightforward data visualisations that allow for an easy assessment of the mortality risk associated with HGS measured in kg in conjunction with patients' age and body height. All statistical analyses were conducted using R.

## Patient and public involvement

There was no direct patient or public involvement in this research, as the analysis is based on secondary observational data.

## RESULTS

### HRs of mortality by HGS group and gender

Results from the Cox proportional hazard models suggest that survival starts decreasing just below the group-specific average HGS, namely in groups weak 1 and 2. The HR of mortality by gender and across standardised HGS groups are presented in table 2, rows 5 and 9 (see also online supplemental table A1). The comparison of the male reference group [0.0 SD, 0.5 SD) with three weaker groups suggests that men's mortality risk increases in a monotonous fashion with each weaker group. The risk increases substantially when the standardised HGS falls into the weak 1 group [−0.5 SD, 0.0 SD) with an HR of 1.67 (95% CI 1.23 to 2.26). Subsequently, the HR further increases to 2.02 (95% 1.49 CI 2.75) in the weak 2 group and to 2.40 (95% CI 1.77 to 3.26) in the weak 3 group. The weakest group [−3.0 to −2.0) comprised only a small fraction of the sample (2.9%) and showed an HR of 2.34 with a large CI (1.40 to 3.93). The reference group and the strong group [0.5 SD, 3.0 SD) were similar in terms of mortality risks (men: HR 0.93, 95% CI 0.69 to 1.25;

women: 0.90, 95% CI 0.66 to 1.22), indicating that there are no survival benefits associated with above-average HGS. The pattern of results for women is similar (table 2, row 9) but involved a somewhat smaller HR that increased from 1.32 (95% CI 0.96 to 1.82) in weak 1 to 1.65 (95% CI 1.20 to 2.28) in weak 2, 1.85 (95% CI 1.34 to 2.55) in weak 3, and 3.03 (95% CI 1.83 to 5.04) in weak 4. Sensitivity analyses controlling for educational attainment yielded virtually identical results (see online supplemental table A1). Survival curves and numbers at risk are shown in online supplemental figure A2.

### Differences in RLE by HGS thresholds and gender

Similar to the results from the Cox proportional hazard models, RLE shows how small negative deviations from the reference group have substantial impact on life expectancy for both genders. Table 3 provides RLE and threshold values for each HGS group for two reference ages, namely 60 and 70 years, with average height being 175 cm for men and 163 cm for women. For example, men aged 60 years in the weak 1 group [−0.5 SD, 0.0 SD) are estimated to have an RLE of 18.4 years — about 3.0 years less than the reference group (21.4 years). Their counterparts in the weak 3 group [−2.0 SD, −1.0 SD) have an estimated RLE of 16.2 years—5.2 years less than the reference group. Group differences in RLE among women are somewhat smaller but still substantial. The weak 3 group is, for example, estimated to have an RLE of 20.5 at age 60 years, which is 3.3 years lower than that of the reference group (23.8 years).

### Group-specific HGS thresholds in kg

Table 3 also provides group-specific threshold values in kg for two reference ages with average height, which further highlight how even small negative deviations in HGS from the norm have a substantial impact on survival. For a 60-year-old man with an average height of 175 cm, for example, those with HGS of less than 45.9 kg fall into the weak 1 group, which already shows a substantially elevated mortality risk compared with the age-specific reference group, which has an HGS of 45.9–49.4 kg. Thus, practitioners should be looking into underlying risk factors when the HGS falls short of the norm by a few kg but should certainly be concerned when the HGS is 0.5 SD below the group-specific threshold.

Notably, for 70-year-old men of average height, these threshold values are substantially lower, for example, the threshold for weak 1 is 41.2 kg instead of 45.9 kg for the individuals aged 60 years, which underlines the claim that group-specific threshold values are required instead of absolute threshold values in kg when assessing health vulnerabilities and the risk of death.

### Using the HGS thresholds to detect increasing mortality risk early on

Findings from the Cox proportional hazard model and also the RLE show that practitioners should start being

**Table 3** Threshold values in kg and remaining life expectancy (RLE) for average height (175 cm for men and 163 cm for women) and two reference ages (60 and 70 years) by gender and st_HGS group

| Groups | Thresholds | Age 60 years | | Age 70 years | |
|---|---|---|---|---|---|
| | | Range in kg | RLE | Range in kg | RLE |
| **Men** | | | | | |
| Strong: | [0.5 SD, 3.0 SD) | [49.4, max) | 21.8 | [44.8, max) | 15.4 |
| Reference: | [0.0 SD, 0.5 SD) | [45.9, 49.4) | 21.4 | [41.2, 44.8) | 15.1 |
| Weak 1: | [−0.5 SD, 0.0 SD) | [42.3, 45.9) | 18.4 | [37.7, 41.2) | 13.0 |
| Weak 2: | [−1.0 SD, −0.5 SD) | [38.7, 42.3) | 17.3 | [34.1, 37.7) | 12.1 |
| Weak 3: | [−2.0 SD, −1.0 SD) | [31.6, 38.7) | 16.2 | [27.0, 34.1) | 11.3 |
| Weak 4: | [−3.0 SD, −2.0 SD) | [min, 31.6) | 16.5 | [min, 27.0) | 11.5 |
| **Women** | | | | | |
| Strong: | [0.5 SD, 3.0 SD) | [30.7, max) | 24.3 | [27.7, max) | 16.5 |
| Reference: | [0.0 SD, 0.5 SD) | [28.4, 30.7) | 23.8 | [25.4, 27.7) | 16.2 |
| Weak 1: | [−0.5 SD, 0.0 SD) | [26.0, 28.4) | 22.4 | [23.0, 25.4) | 15.2 |
| Weak 2: | [−1.0 SD, −0.5 SD) | [23.7, 26.0) | 21.2 | [20.6, 23.0) | 14.4 |
| Weak 3: | [−2.0 SD, −1.0 SD) | [18.9, 23.7) | 20.5 | [15.9, 20.6) | 13.9 |
| Weak 4: | [−3.0 SD, −2.0 SD) | [min, 18.9) | 17.5 | [min, 15.9) | 11.7 |

HGS, handgrip strength; SD, standard deviation.

concerned about underlying and potentially undiagnosed risks when the HGS is just below that of the reference group, as mortality risk increases already for the group [−0.5 SD, 0.0 SD). In figure 1, we provide visualisations that allow for a straightforward assessment of survival based on our models. The graph directly links HGS in kg to the HGS

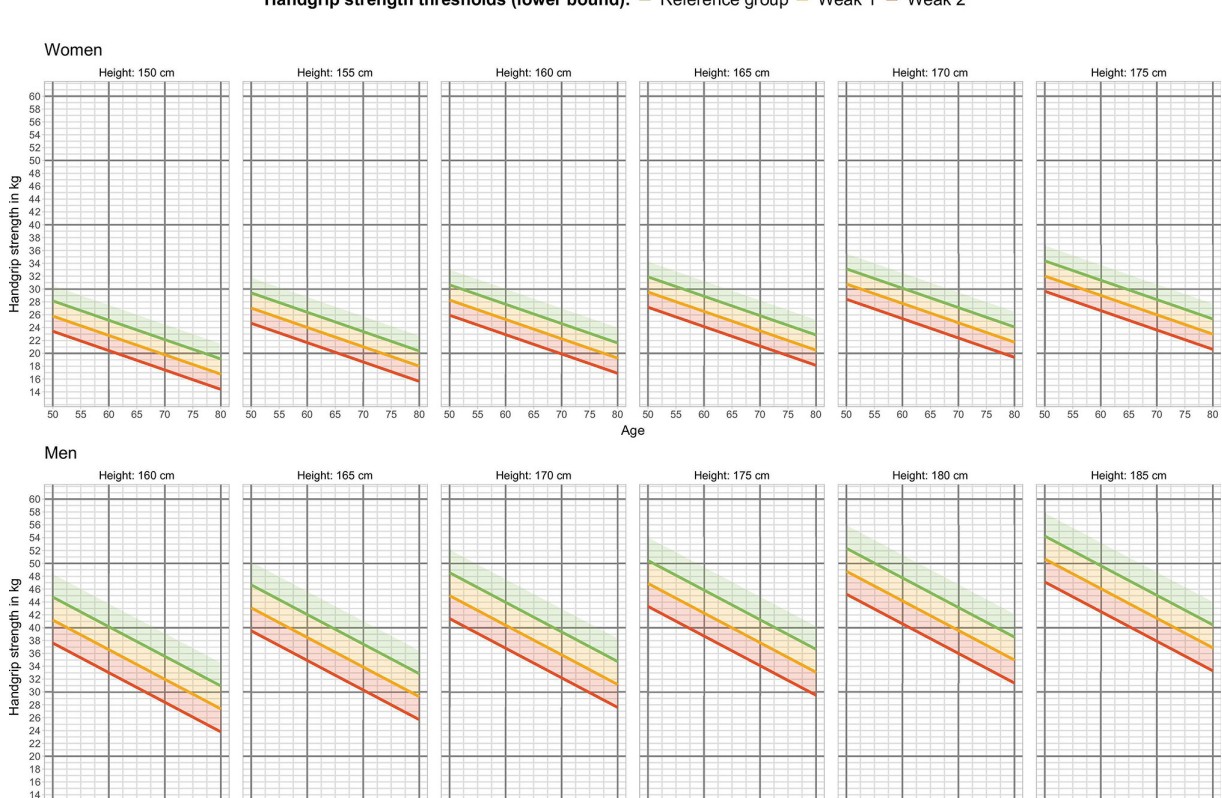

**Figure 1** Model-based HGS thresholds (lower bound) in kg by gender, age and body height. The reference group refers to HGS at or slightly above the mean [0.0 SD, 0.5 SD), weak 1 to HGS up to 0.5 SD below the mean [−0.5 SD, 0.0 SD) and weak 2 to between a half and a full SD below the mean [−1.0 SD, −0.5 SD). HGS, handgrip strength; SD, standard deviation.

thresholds most relevant for an early detection of increased mortality risk, namely the reference group, the weak 1 group and the weak 2 group.

Age-specific thresholds are provided for a set of gender–height combinations, given that differences in HGS (in kg) between genders and height groups are not informative for mortality assessments. The graph shows, for example, that a man aged 70 years with a height of 170 cm and a measured HGS of 40 kg would be above the threshold for the reference group (green line), and this would not lead to concerns regarding increased mortality risks. A taller man of 175 cm, however, with same age and same measured HGS would fall into the weak 1 group (above orange line), suggesting that a more thorough health screening and health-enhancing clinical interventions may be needed.

## DISCUSSION

This study aimed to provide HGS thresholds for the general population to enable early detection of increased mortality risks related to muscle weakness. The contributions made have wide applicability, given that the cut-off points provided in this paper, along with a visual illustration of them, can be routinely implemented in medical practice and that their link to RLE can be communicated to patients much more easily than previously reported HR. These thresholds can thus be used as a screening tool to identify patients who would benefit from further assessments, healthcare interventions or lifestyle changes.

The majority of clinical cut-off values for defining a weak grip are given by the HGS measured in kg, separately by gender, but irrespective of age and body height. To the best of our knowledge, this is the first study to provide HGS thresholds based on representative data that take into account both the inherently non-informative increase in muscle strength in conjunction with body height and the decrease in HGS with age. We advocate the use of z-standardised HGS in clinical practice, based on our results confirming that HGS varies substantially with gender, age and height, which calls for reference points that differentiate between these dimensions.

Our results show that deviations from the reference group matter only for HGS below the group-specific 'norm' (reference group), whereas superior HGS has no beneficial effect on survival. Moreover, we find that survival decreases monotonically with below-average HGS, preventing the use of log-rank or similar methods for identifying optimal cut-off points. Previous evidence regarding the shape of the association between HGS and mortality is mixed. A recent meta-analysis concludes that the relationship is linear,[5] while other studies suggest that high levels of HGS do not provide additional protection for mortality compared with medium levels,[27] thus supporting our findings.

From our findings, it is clear that HGS, when standardised for gender, age and body height, is a precise biomarker of (non-)healthy ageing and indicative of an increased risk of death, even if it deviates by only half an SD from the norm

(ie, the respective reference group). For example, when comparing men aged 50–80 years in the reference group [0.0 SD, 0.5 SD) with men of the same age and body height but with slightly weaker HGS [−0.5 SD, 0.0 SD), the latter were shown to be 67% more likely to die earlier (HR 1.67, 95% CI 1.23 to 2.26). Notably, this HR is similar to prior work that has used substantially lower HGS thresholds (ie, that included much weaker individuals in the risk group).[28] Using HRS data, Duchowny[29] defined muscle weakness as <35 kg for Caucasian men and <22 kg for women and showed it to be associated with a 50% greater risk of death over the follow-up period of 9 years (HR: 1.52, 95% CI 1.15 to 1.47). Again, these absolute thresholds are substantially higher than the ones suggested in our study. In the present paper, differences in HGS in kg between the group-specific reference and the group weak 1, which has an HR of 1.67, are quite small, highlighting once again how exact HGS measurements are as a screening tool. As shown in table 2, the difference in the average HGS for men between the reference group and the group weak 1 is about 4 kg and even smaller for women. This is also clear from figure 1, where the threshold lines of each risk group are very tight.

Most of the limitations of this study are data driven. First, observations were made only for Caucasian individuals due to the small number of observations from other races; this is relevant given that previous studies have found substantial differences in reference HGS by race.[23] Second, the frailest of the survey participants may not be able to perform the HGS test. Previous studies have shown that these individuals also have naturally higher mortality and their non-response in this case could lead to an underestimation of mortality in our sample.[30] Our analysis is thus aimed at patients above a certain strength- and health-related threshold, who are able to perform the HGS test when consulting a practitioner and are, analogously, able to perform the HGS test when participating in a survey.

Despite its data limitations, this study has confirmed that muscle strength is a powerful and precise predictor of survival and that HGS is a quick and inexpensive way of assessing reductions in RLE in clinical practice, as long as inherent variations by gender, age and body height are considered in the cut-off points applied. Our evidence points towards a threshold value for defining a critically weak handgrip that is much higher than that assumed in gerontological research to date. Survival decreases just below the gender-specific, age-specific and height-specific HGS norms, which is why we suggest a cut-off point of 0.5—or even smaller—to detect patients with an increased mortality risk early on.

**Contributors** SSc and NS designed the study; SSc was responsible for data curation and the statistical analyses; NS wrote the original draft, and SSp was responsible for reviewing and editing the manuscript and for data visualisation. All authors have verified the underlying data and are responsible for the overall content. The corresponding author attests that all listed authors meet authorship criteria and that no others meeting the criteria have been omitted and acts as a guarantor.

**Funding** Open access funding was provided by the University of Vienna.

**Disclaimer** The funder of the study had no role in study design, data analysis, data interpretation, writing of the manuscript; or decision to submit the manuscript for publication. All authors are independent from the funder and had full access to all of the data in the study and thus can take responsibility for the integrity of the data and the accuracy of the data analysis.

**Competing interests** None declared.

**Patient and public involvement** Patients and/or the public were not involved in the design, or conduct, or reporting, or dissemination plans of this research.

**Patient consent for publication** Not applicable.

**Ethics approval** This manuscript is based on secondary data from the Health and Retirement Study, which has been conducted according to the Declaration of Helsinki and was approved by the University of Michigan Institutional Review Board (UM Health Sciences/Behavioral Sciences IRB Protocol: HUM00061128 approved through 10/18/2018 Associated protocols: HUM00056464, HUM00002562, HUM00074501, HUM00079949, HUM00080925, HUM00085942, HUM00099822, HUM00103072, HUM00106904, HUM00122335, REP00000046). Participants gave informed consent to participate in the study before taking part.

**Provenance and peer review** Not commissioned; externally peer reviewed.

**Data availability statement** Data may be obtained from a third party and are not publicly available. The study used anonymised secondary data that are made available to all interested researchers on request from the Health and Retirement Study.

**ORCID iDs**
Sergei Scherbov http://orcid.org/0000-0002-0881-1073
Sonja Spitzer http://orcid.org/0000-0002-2114-7947
Nadia Steiber http://orcid.org/0000-0002-9425-8840

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
