## [Reviewer comments · BMJ Open]

ARTICLE DETAILS

TITLE (PROVISIONAL)	Thresholds for clinical practice that directly link handgrip strength to remaining years of life: Estimates based on longitudinal observational data
AUTHORS	Scherbov, Sergei; Steiber, Nadia; Spitzer, Sonja

VERSION 1 – REVIEW

REVIEWER	Xu, Zhijie Zhejiang University School of Medicine Second Affiliated Hospital, Department of General Practice
REVIEW RETURNED	22-Dec-2021

GENERAL COMMENTS	This study provided HGS thresholds for the general population, which could be a screening tool. The overall paper is well articulated and is suitable for publication in the current form. However, there are some clerical errors in the manuscript. Please complete a thorough proofread of the text and correct any spelling and grammar errors.
---

REVIEWER	Rębacz-Marón, Ewa Uniwersytet Szczeciński, Faculty of Biology
REVIEW RETURNED	23-Dec-2021

GENERAL COMMENTS	Very important topic ! Dynamometry is non-invasive, cheap to use and uncomplicated to measure. HGS is useful in recovery and also in gerontology. The article is very labor intensive. The authors did a lot of statistical calculations and collected extensive data beforehand. In my opinion, a very important sentence: „Our evidence points towards a threshold value for defining a critically weak handgrip that is much higher compared to what has been assumed in gerontological research so far.”
--

REVIEWER	Lee, Yunhwan Ajou University School of Medicine and Graduate School of Medicine, Preventive Medicine and Public Health
REVIEW RETURNED	04-Feb-2022

GENERAL COMMENTS	This study suggests new thresholds for the handgrip strength (HGS) adjusted for age and height to predict mortality in people aged 50 to 80 years. Even a slightly lower value in the standardized handgrip strength indicates increased mortality risk. Presentation of the risk regarding remaining life expectancy (RLE) may better communicate information to patients. Use of national data and a novel approach to determine HGS thresholds are
---

	commendable. I have several comments for the authors to consider. p. 5, In 25: The use of nationally representative data adds to the generalizability of the results. However, the authors should provide more information about the sample used in the analysis. Notwithstanding selecting the subgroups from the total study population, attrition during follow-up would have reduced the sample size. A flowchart may serve to elucidate the sample selection process. Also, a sensitivity analysis might be warranted to account for the missing data. p. 7, In 44: As the author stated, RLE provides an estimate of the segmented life expectancy for different age groups, for example, from 60 to 90. I am not sure that I understand this correctly, but wouldn't this pose a problem for estimating the age-specific risk? Also, because the cut-off points are taken at 5-year intervals, wouldn't this oversimplify the estimate? p. 11, In 31: A potential threat to internal validity, that is, selection bias, is mentioned. It appears that it would be important to consider health conditions in determining thresholds as frailty and other geriatric conditions would hinder and influence the measurement. Figure 1: It is unclear what the colors represent unless referring to the text. Please provide captions.
--	--

VERSION 1 – AUTHOR RESPONSE

REVIEWER 1

Dr. Zhijie Xu, Zhejiang University School of Medicine Second Affiliated Hospital

This study provided HGS thresholds for the general population, which could be a screening tool. The overall paper is well articulated and is suitable for publication in the current form. However, there are some clerical errors in the manuscript. Please complete a thorough proofread of the text and correct any spelling and grammar errors.

Response: Thank you for the positive evaluation of our manuscript. Based on this comment, we have now hired a professional language editor who corrected any spelling and grammar errors.

REVIEWER 2

Dr. Ewa Rębacz-Marón, Uniwersytet Szczeciński

Very important topic!

Dynamometry is non-invasive, cheap to use and uncomplicated to measure.

HGS is useful in recovery and also in gerontology.

The article is very labor intensive. The authors did a lot of statistical calculations and collected extensive data beforehand.

In my opinion, a very important sentence: "Our evidence points towards a threshold value for defining a critically weak handgrip that is much higher compared to what has been assumed in gerontological research so far."

Response: We thank the Reviewer for this positive comment. Based on the editor's suggestion, we have removed the Sentence "Our evidence points towards a threshold value for defining a critically weak handgrip that is much higher compared to what has been assumed in gerontological research so far" from the "Strength and limitations" section, and moved it to the "Discussion" section instead, where it is now concluding the paper.

REVIEWER 3

Dr. Yunhwan Lee, Ajou University

This study suggests new thresholds for the handgrip strength (HGS) adjusted for age and height to predict mortality in people aged 50 to 80 years. Even a slightly lower value in the standardized handgrip strength indicates increased mortality risk. Presentation of the risk regarding remaining life expectancy (RLE) may better communicate information to patients. Use of national data and a novel approach to determine HGS thresholds are commendable. I have several comments for the authors to consider.

1. p. 5, ln 25: The use of nationally representative data adds to the generalizability of the results. However, the authors should provide more information about the sample used in the analysis. Notwithstanding selecting the subgroups from the total study population, attrition during follow-up would have reduced the sample size. A flowchart may serve to elucidate the sample selection process. Also, a sensitivity analysis might be warranted to account for the missing data.

Response: Based on this valid suggestion, we have now added a chart that visualises the most important steps of the sample construction to the Appendix.

Furthermore, we agree with the Reviewer that, in general, attrition during follow-up would reduce the sample size and potentially bias estimates. Unregistered attrition due to death is, however, known

to be very small in the Health and Retirement Study. Moreover, those who drop out of the sample before dying can be tracked regardless and matched with the National Death Index. Consequently, it is possible to identify the date of death for each observation, even if they leave the survey early, which is a major advantage of working with the Health and Retirement Study. Since our analysis focuses mainly on individuals' mortality, survey attrition is thus not a concern for our paper. Based on this comment, we have now added the following sentence clarifying this to the manuscript on page 5: "Participants can be linked to the National Death Index even if they leave the study early on, making attrition a minor concern for this analysis".

2. p. 7, ln 44: As the author stated, RLE provides an estimate of the segmented life expectancy for different age groups, for example, from 60 to 90. I am not sure that I understand this correctly, but wouldn't this pose a problem for estimating the age-specific risk? Also, because the cut-off points are taken at 5-year intervals, wouldn't this oversimplify the estimate?

Response: Thank you for this question. Segmented life tables are computed based on estimated age-specific risks and not vice versa. Hence, restricting our analysis to those aged 60 to 90 – i.e. computing segmented life tables – does not affect the estimated age-specific risks. The advantage of employing segmented life tables is that they allow us to address the small number of observations at older ages, which could otherwise cause unreliable estimates. In other words, the life expectancies presented in our article are based on the sum of person-years lived in the age interval 60 to 90. Similarly, the 5-year intervals do not bias or oversimplify the estimates, but instead make them more reliable, by reducing noise. We hope this explanation has made this demographic approach clearer, and we are happy to clarify any additional concerns regarding the construction of life tables.

3. p. 11, ln 31: A potential threat to internal validity, that is, selection bias, is mentioned. It appears that it would be important to consider health conditions in determining thresholds as frailty and other geriatric conditions would hinder and influence the measurement.

Response: This paper aims at providing cut-off points for handgrip strength that enable practitioners to detect patients with an increased mortality risk early on. Age-specific thresholds – like those in Figure 1 – are provided for a set of gender-height combinations, since differences in handgrip strength between genders and height-groups are not informative for mortality assessments.

In contrast, frail individuals may have, on average, lower handgrip strength, precisely, because they have a higher risk of dying, which is why we do not consider frailty and other geriatric conditions in our cut-off points. Moreover, if we were to consider health conditions such as frailty when constructing the cut-off points – i.e. have separate cut-off points for various levels of frailty – a patient's level of frailty would have to be determined before her handgrip strength is measured, which would complicate the medical practitioners' assessment of mortality risks. We thus refrain from considering additional health dimensions for the thresholds.

We do, however, acknowledge that our cut-off points aim at patients that are able to conduct the handgrip strength test, and have now added the following sentence to the study limitations on page 11: “Our analysis is thus aimed at patients above a certain strength- and health-related threshold, who are able to perform the HGS test when consulting a practitioner and, analogously, able to perform the HGS test when participating in a survey.”

4. Figure 1: It is unclear what the colors represent unless referring to the text. Please provide captions.

Response: Thank you for this comment. The legend for Figure 1 can be found in the upper area of the graph. We have now increased the size of that legend and, in addition, extended the existing caption, which now explains the handgrip strength thresholds in more details. We hope the Reviewer finds this improvement satisfactory.

VERSION 2 – REVIEW

REVIEWER	Lee, Yunhwan Ajou University School of Medicine and Graduate School of Medicine, Preventive Medicine and Public Health
REVIEW RETURNED	09-Jun-2022
GENERAL COMMENTS	The authors have addressed all my concerns. I have no further comments.